

# Anti-BIRC5 autoantibody serves as a valuable biomarker for diagnosing AFP-negative hepatocellular carcinoma

Qing Li[1,2,3,4], Haiyan Liu[1], Han Wang[1], Wenzhuo Xiong[3,4], Liping Dai[1,3,4], Xiuzhi Zhang[1,3,4,5], Peng Wang[1], Hua Ye[1], Jianxiang Shi[1,4], Zhihao Fang[1] and Keyan Wang[1,3,4]

[1] Henan Institute of Medical and Pharmaceutical Sciences, Zhengzhou University, Zhengzhou, Henan, China
[2] Henan Key Laboratory of Tumor Epidemiology and State Key Laboratory of Esophageal Cancer Prevention & Treatment, Zhengzhou University, Zhengzhou, Henan, China
[3] School of Basic Medical Sciences, Academy of Medical Science, Zhengzhou University, Zhengzhou, Henan, China
[4] Henan Key Medical Laboratory of Tumor Molecular Biomarkers, Zhengzhou, China
[5] Department of Pathology, Henan Medical College, Zhengzhou, Henan, China

Corresponding author
Keyan Wang,
keyanwang@zzu.edu.cn

## ABSTRACT

**Background:** Autoantibodies targeting tumor-associated antigens (TAAbs) have emerged as promising biomarkers for early cancer detection. This research aimed to assess the diagnostic capacity of anti-BIRC5 autoantibody in detecting AFP-negative hepatocellular carcinoma (ANHCC).

**Methods:** This research was carried out in three stages (discovery phase, validation phase, and evaluation phase) and included a total of 744 participants. Firstly, the anti-BIRC5 autoantibody was discovered using protein microarray, exhibiting a higher positive rate in ANHCC samples (ANHCCs) compared to normal control samples (NCs). Secondly, the anti-BIRC5 autoantibody was validated through enzyme-linked immunosorbent assay (ELISA) in 85 ANHCCs and 85 NCs from two clinical centers (Zhengzhou and Nanchang). Lastly, the diagnostic usefulness of the anti-BIRC5 autoantibody for hepatocellular carcinoma (HCC) was evaluated by ELISA in a cohort consisting of an additional 149 AFP-positive hepatocellular carcinoma samples (APHCCs), 95 ANHCCs and 244 NCs. The association of elevated autoantibody to high expression of BIRC5 in HCC was further explored by the database from prognosis, immune infiltration, DNA methylation, and gene mutation level.

**Results:** In the validation phase, the area under the ROC curve (AUC) of anti-BIRC5 autoantibody to distinguish ANHCCs from NCs in Zhengzhou and Nanchang centers was 0.733 and 0.745, respectively. In the evaluation phase, the AUCs of anti-BIRC5 autoantibody for identifying ANHCCs and HCCs from NCs were 0.738 and 0.726, respectively. Furthermore, when combined with AFP, the AUC for identifying HCCs from NCs increased to 0.914 with a sensitivity of 77.5% and specificity of 91.8%. High expression of BIRC5 gene is not only correlated with poor prognosis of HCCs, but also significantly associated with infiltration of immune cells, DNA methylation, and gene mutation.

**Conclusion:** The findings suggest that the anti-BIRC5 autoantibody could serve as a potential biomarker for ANHCC, in addition to its supplementary role alongside

AFP in the diagnosis of HCC. Next, we can carry out specific verification and explore the function of anti-BIRC5 autoantibody in the occurrence and development of HCC.

## INTRODUCTION

Primary liver cancer (PLC) ranks as the sixth most prevalent malignancy globally, representing the second highest cause of male cancer deaths and the third most common reason for mortality related to cancer (*Sung et al., 2021*). AFP is the only frequently utilized serological biomarker in HCC detection and prognostic monitoring, but its sensitivity and specificity are limited (*Pang et al., 2008*). At a threshold level of 20 ng/ml, the sensitivity of AFP was observed to vary between 40% and 60%, while its specificity ranged from 80% to 90% (*Yang et al., 2019*). Patients with AFP-negative hepatocellular carcinoma (ANHCC) often present with small tumor sizes or are in the early stages of the disease, exhibiting minimal clinical symptoms. Furthermore, their imaging characteristics closely resemble those of benign lesions, posing a challenge for detection using conventional imaging modalities such as ultrasound (*Wang & Zhang, 2020*). Therefore, there is an urgent need for identifying novel serological or noninvasive markers to accurately diagnose ANHCC patients and improve prognosis.

Tumor-associated antigens (TAAs) are proteins implicated in tumorigenesis, capable of initiating a cascade of immunological reactions leading to the production of autoantibodies against these tumor-associated antigens (*Liu et al., 2011*). The utilization of TAAs and TAAbs systems as biomarkers has recently been extensively investigated for cancer detection or monitoring treatment efficacy. It has been demonstrated that TAAbs are consistently absent or present at extremely low levels in healthy individuals and non-cancerous conditions, whereas their concentration is significantly elevated in tumors (*Old & Chen, 1998*; *Tan & Zhang, 2008*; *Zhang & Tan, 2010*). Although TAAs currently serve as the most commonly utilized biomarkers for early cancer detection, their circulating half-life is limited due to rapid degradation or elimination (*Anderson & LaBaer, 2005*). TAAbs may arise early in carcinogenesis when TAAs are present in precancerous or malignant lesions. The immune system can efficiently amplify and memorize the immune response to these antigens, hereby rendering TAAb a promising cancer biomarker (*Zhang et al., 2020*).

BIRC5, the tiniest member within the family of inhibitor of apoptosis protein (IAP), comprises 142 amino acid residues (*Hunter, LaCasse & Korneluk, 2007*; *Schimmer, 2004*). It is present in a physiological state as a functional homodimer. In contrast to other IAPs, BIRC5 encompasses a BIR domain spanning from amino acid position 15 to amino acid position 87 (*Chantalat et al., 2000*; *Verdecia et al., 2000*). BIRC5 not only hinders the process of programmed cell death but also facilitates cellular replication. When there is an

excessive expression of BIRC5, it obstructs apoptosis through various mechanisms, thereby promoting abnormal cell growth and the development of malignancies (*Athanasoula et al., 2014*; *Salzano et al., 2014*; *Wang et al., 2015*; *Zhang et al., 2015*). BIRC5 exhibits a significantly elevated level in nearly all types of cancer (*Athanasoula et al., 2014*; *Necochea-Campion et al., 2013*). Immunohistochemical analysis revealed that BIRC5 exhibited a positive rate of 70% in HCC tissues, while it was not detected in adjacent non-cancerous tissues and cirrhotic tissues (*Ito et al., 2000*). Previous studies have predominantly focused on investigating the gene and protein aspects of BIRC5 (*Athanasoula et al., 2014*; *Ito et al., 2000*; *Necochea-Campion et al., 2013*; *Salzano et al., 2014*; *Wang et al., 2015*; *Zhang et al., 2015*). In this study, we identified an elevated occurrence of anti-BIRC5 autoantibody in ANHCCs compared to NCs using protein microarray technology.

Therefore, in order to explore, confirm, and evaluate the diagnostic significance of anti-BIRC5 autoantibody in patients with ANHCC as well as in whole HCC patients, a total of 744 individuals were enrolled for this study and categorized into the discovery, validation and evaluation stages. Following the validation across multiple centers and stages, it was demonstrated that the serum marker of anti-BIRC5 autoantibody exhibits high reliability and accuracy. The impact of BIRC5 gene expression on patient prognosis and its association with infiltrated immune cells, DNA methylation, and gene mutation were further investigated.

## MATERIALS AND METHODS

### Serum samples

A set of 744 human serum samples were examined and divided into three separate stages: the discovery phase, validation phase, and evaluation phase. In the discovery phase, sera form 39 ANHCCs and 47 NCs were included and tested by protein microarray. For the validation phase, a total of 114 samples from Zhengzhou (57 ANHCCs, 57 NCs), as well as 56 samples from Nanchang (28 ANHCCs, 28 NCs) were incorporated to validate the diagnostic performance of anti-BIRC5 autoantibody in ANHCCs. Furthermore, an additional substantial sample set consisting of 488 samples was included in the evaluation phase. This set comprised of 95 ANHCCs, 149 APHCCs, and 244 NCs. In both the validation and evaluation phases, matching by gender and age was ensured between ANHCCs and NCs. The essential information of the participants is detailed in Table 1. The serum specimens were stored in the serum bank of the Tumor Epidemiology Laboratory of Zhengzhou University (Henan, China). Informed consent forms were signed by all subjects, and the study was approved by the Ethics Review Board of Zhengzhou University (ZZURIB2019-001). Diagnosis of HCC patients in this study were based on criteria established in 2017 in China (*Zhou et al., 2018*). Staging of HCC followed the eighth edition of the American Joint Committee on Cancer (AJCC) Cancer Staging Manual (*Chun, Pawlik & Vauthey, 2018*). All NC individuals had no history of liver diseases, autoimmune disease, excessive alcohol consumption, family history of cancer, personal history of cancer, cirrhosis, fatty liver, hepatitis B, or hepatitis C.

**Table 1 Characteristics of participants.**

| | Validation phase | | | | Evaluation phase | | |
|---|---|---|---|---|---|---|---|
| | Cohort 1 | | Cohort 2 | | Cohort 3 | | |
| | ANHCC | NC | ANHCC | NC | ANHCC | APHCC | NC |
| N | 57 | 57 | 28 | 28 | 95 | 149 | 244 |
| Gender, n (%) | | | | | | | |
| Male | 47 (82.5) | 47 (82.5) | 20 (71.4) | 20 (71.4) | 74 (77.9) | 110 (73.8) | 186 (76.2) |
| Female | 10 (17.5) | 10 (17.5) | 8 (28.6) | 8 (28.6) | 21 (22.1) | 39 (26.2) | 58 (23.8) |
| Age, years | | | | | | | |
| Range | 32–79 | 32.5–88 | 38–68 | 36–72 | 42–84 | 26–87 | 34–87.5 |
| Mean ± SD | 54.7 ± 12.2 | 56.7 ± 12.5 | 53.8 ± 7.6 | 54.3 ± 8.8 | 57.6 ± 9.3 | 56.1 ± 10.4 | 56.6 ± 10.1 |
| TNM stage, n (%) | | | | | | | |
| I + II | 20 (35.1) | IA | 0 | IA | 46 (48.4) | 52 (34.9) | IA |
| III + IV | 24 (42.1) | IA | 0 | IA | 39 (41.1) | 90 (60.4) | IA |
| NA | 13 (22.8) | IA | 28 (100) | IA | 10 (10.5) | 7 (4.7) | IA |
| HBV infection, n (%) | | | | | | | |
| Yes | 34 (59.6) | 0 | 7 (25.0) | IA | 59 (62.1) | 124 (83.2) | 0 |
| No | 18 (31.6) | 57 (100) | 2 (7.1) | IA | 32 (33.7) | 23 (15.5) | 244 (100) |
| NA | 5 (8.8) | 0 | 19 (67.9) | IA | 4 (4.2) | 2 (1.3) | 0 |
| Metastasis, n (%) | | | | | | | |
| Yes | 5 (8.8) | IA | 0 | IA | 12 (12.6) | 27 (18.1) | IA |
| No | 52 (91.2) | IA | 0 | IA | 83 (87.4) | 122 (81.9) | IA |
| NA | 0 | 0 | 28 (100) | IA | 0 | 0 | IA |

Note:
HCC, hepatocellular carcinoma; NC, normal control; HBV infection, combined hepatitis B virus; SD, standard deviation; AFP, alpha-fetoprotein; APHCC, AFP ≥ 20 ng/mL; ANHCC, AFP < 20 ng/ml. IA, inapplicable.

### Focused protein microarray

A protein microarray with a specific focus was created, consisting of 154 proteins derived from 138 genes associated with cancer progression, following the methodology described in previous studies (*Ma et al., 2021*; *Wang et al., 2020*). The detailed procedure for the detection of multiple autoantibodies including autoantibody to BIRC5 in ANHCC and NC sera was outlined in our recent study (*Ma et al., 2021*; *Wang et al., 2020*). Signal-to-noise ratio (SNR) values were used to represent the levels of anti-BIRC5 autoantibody.

### Enzyme-linked immunosorbent assay

The enzyme-linked immunosorbent assay (ELISA) utilized the BIRC5 recombinant protein, which was procured from Cloud-Clone Corporation (Wuhan, China). The concentration of the coated protein was repeatedly tested and determined to be 0.5 μg/ml. The sera samples underwent a dilution of 1:100. A comprehensive protocol was provided in our previously published research articles (*Ma et al., 2021*; *Wang et al., 2020*). In summary, the TMB color rendering technique was utilized during the validation and assessment stage to identify autoantibody targeting BIRC5 (*Dai et al., 2014*). For each test, two empty controls and eight human serum samples with consistent values were included
on every 96-well plate to account for background adjustment and ensure the normalization of OD values across various plates.

## Statistical analysis

Statistics was analyzed using GraphPad Prism 8.0 software and IBM SPSS 21.0 software. The receiver operating characteristics (ROC) curve analysis was used to evaluate the diagnostic value. The sensitivity, specificity, PPV, and NPV and accuracy were calculated to evaluate the validity and reliability. Non -parametric statistical tests, namely the Mann-Whitney U test and Kruskal-Wallis H-test, were employed to assess the significance of SNR or optical density (OD) values across two or multiple groups. The determination of the cut-off value was based on the 90th percentile of Nanchang normal controls. To evaluate the significance of autoantibody frequency among different groups within each dataset, Pearson's chi-squared test was utilized. Statistical significance was considered for *P*-values below 0.05.

# RESULT

## Performance of anti-BIRC5 autoantibody in ANHCC during the discovery phase

The flow chart of this study is shown in Fig. 1. Results from the focused protein microarrays demonstrate that the SNR was relatively higher in ANHCCs than in NCs, as depicted in Figs. 2A and 2B. In ANHCCs, the positive rate of anti-BIRC5 autoantibody was found to be 38.5%, whereas it was only 10.6% in NCs, with a statistically significant difference between the two groups ($\chi^2 = 9.245$, $P = 0.002$) (Fig. 2C). Notably, the SNR level of anti-BIRC5 autoantibody in ANHCCs was significantly elevated when compared to that observed in NCs ($P = 0.0005$) (Fig. 2D). Setting the cutoff value at 1.85 based on the 90th percentile of NC samples revealed that the sensitivity and specificity of anti-BIRC5 autoantibody for distinguishing ANHCC from NC were determined as 38.5% and 89.4%, respectively, while achieving an AUC value of 0.716 (Fig. 2E).

## Anti-BIRC5 autoantibody was validated in a multicenter study

In Fig. 3, The OD value of ANHCC group was apparently greater than that of NC group in Zhengzhou ($P < 0.001$) (Fig. 3A). The AUC was determined to be 0.733 (Fig. 3B). The positive rate of anti-BIRC5 autoantibody in group ANHCC was significantly higher than that in group NC ($\chi^2 = 8.732$, $P = 0.003$) (Fig. 3C). In Nanchang, the scatter plot revealed a significantly higher titer of anti-BIRC5 autoantibody in ANHCC compared to NC ($P = 0.0014$) (Fig. 3D), with an AUC of 0.745 (Fig. 3E). By setting the cut off value at 0.351 based on the 90% percentile of Nanchang NCs, the positive rate of anti-BIRC5 autoantibody in ANHCCs was 39.3%, which was significantly higher than that in NCs ($\chi^2 = 15.244$, $P < 0.0001$), as shown in Fig. 3F. The evaluating indicators such as sensitivity, specificity, Positive predictive value (PPV), Negative predictive value (NPV), and accuracy were consistent between clinical center in Zhengzhou and Nanchang, indicating stable and universally applicable diagnostic performance for anti-BIRC5 autoantibody (Table 2).

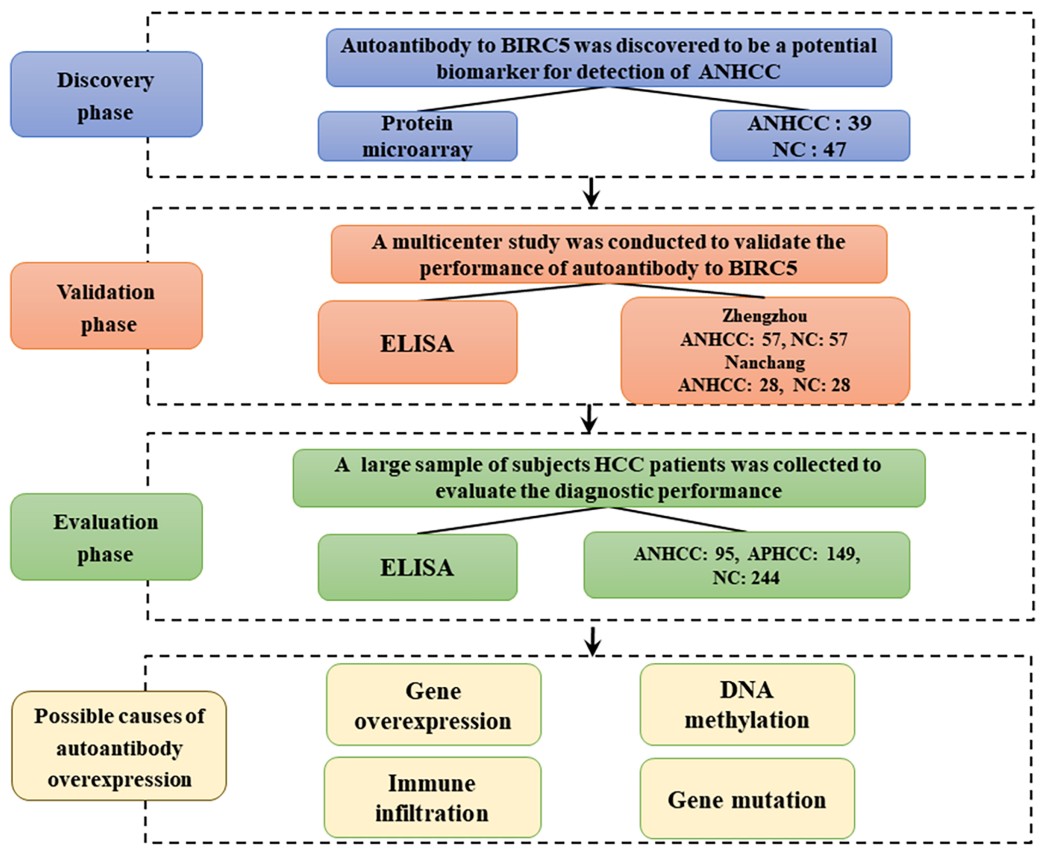

**Figure 1 Work flow chart of the research.** This study was mainly divided into discovery phase, validation phase, and evaluation phase. HCC, hepatocellular carcinoma; NC, normal control; ANHCC, AFP negative hepatocellular carcinoma; APHCC, AFP positive hepatocellular carcinoma; APHCC, AFP ≥ 20 ng/mL; ANHCC, AFP < 20 ng/ml.                 

## Evaluate the diagnostic performance of anti-BIRC5 autoantibody and its association with AFP for whole HCC in a large sample size

In Fig. 4, the AUCs of anti-BIRC5 autoantibody in ANHCC and APHCC are 0.738 and 0.718 (Figs. 4A and 4B). The positive rate between the two groups did not exhibit any significant variance (Fig. 4C). It can be observed from Fig. 4D that the AUC of AFP was 0.844. When combining anti- BIRC5 autoantibody with AFP to diagnose the whole HCC from NC, the AUC reached 0.914, the sensitivity increased to 77.5%, and the specificity was 91.8% (Table 3). We further investigated whether there was a correlation between anti-BIRC5 autoantibody and AFP, however, as shown in Fig. 4E, no such correlation was found ($P = 0.3684$). The Venn diagram illustrating the diagnosis of HCC patients confirmed by analyzing both AFP and anti-BIRC5 autoantibody revealed that an additional 33 cases could be detected when combining these two biomarkers together (Fig. 4F), thereby suggesting that anti-BIRC5 autoantibody plays a complementary role alongside AFP in the diagnosis of HCC.

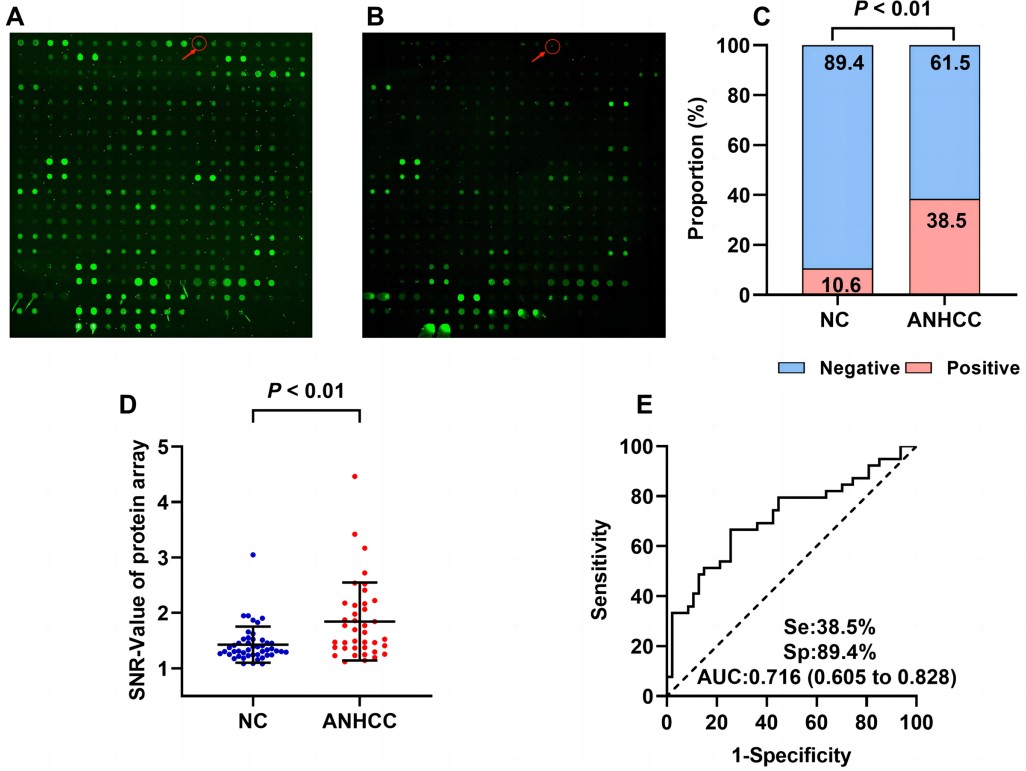

**Figure 2 Anti-BIRC5 autoantibody was discovered by using protein microarrays.** The arrows in (A) and (B) point to the anti-BIRC5 autoantibody microarray scan results in ANHCC and NC. Positive rates (C) and scatter plot of SNR (D) of anti-BIRC5 autoantibody in different groups (the cutoff value was 1.85 determined by the 90% percentile of NC). (E) The ROC of anti-BIRC5 autoantibody in ANHCC.

## Diagnostic performance of anti-BIRC5 autoantibody in different subgroups of HCC

The diagnostic performance of anti-BIRC5 autoantibody was demonstrated in different subgroups (Fig. 5 and Table 4). The sensitivity ranges from 28.2% to 35.7%, while the AUC varies between 0.718 and 0.754. Similar sensitivity is observed across different subgroups when assuming the same cut-off value, suggesting consistency in diagnostic performance for anti-BIRC5 autoantibody across subgroups.

## High expression of BIRC5 in patients with HCC is associated with poor prognosis

The abnormal expression of tumor-associated antigens in malignant tumors triggers an immune response, leading to the production of a majority of tumor-associated autoantibodies. The database (https://www.aclbi.com/static/index.html#/) was utilized to investigate the correlation between BIRC5 gene expression and patient survival. Figure 6A illustrates that an elevation in BIRC5 gene expression is associated with increased mortality rates among patients. Moreover, Fig. 6B demonstrates that a HR > 1 designates this gene as a risk factor, indicating that higher levels of BIRC5 expression are linked to shorter survival times and poorer prognosis for patients. Additionally, Fig. 6C showcases ROC curves

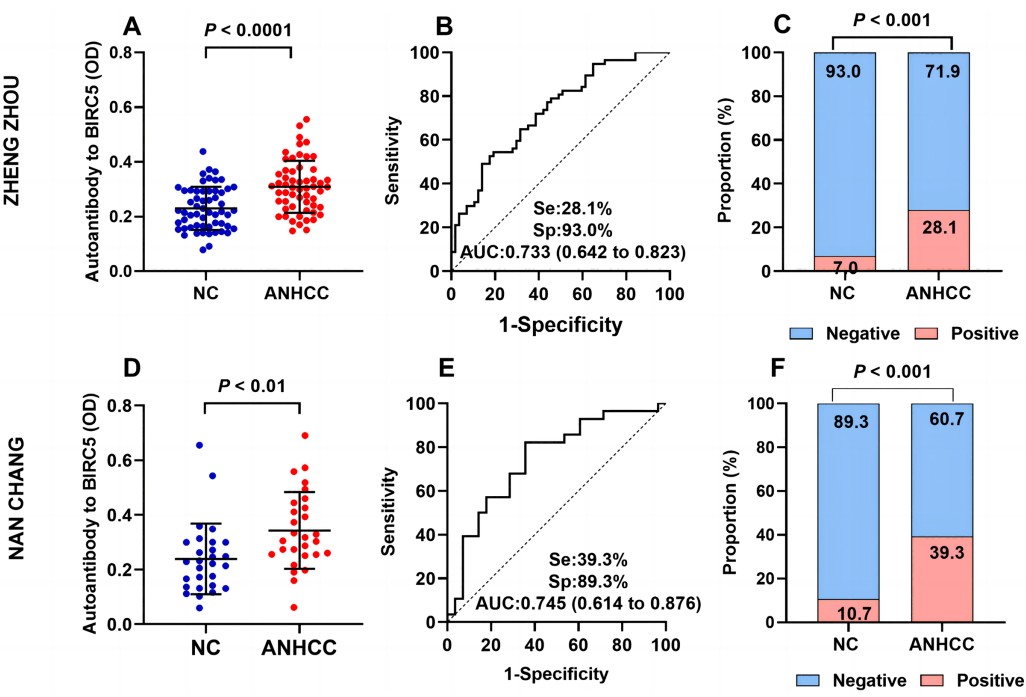

**Figure 3 Multi-center validation of anti-BIRC5 autoantibodies.** (A–C) Validation cohort samples in Zhengzhou. (A) Scatter plot of ANHCC and NC. (B) The ROC of anti-BIRC5 autoantibody in ANHCC. (C) The bar chart of positive and negative rates in the ANHCC and NC. (D–F) Validation cohort samples in Nanchang. (D) Scatter plots of ANHCC and NC. (E) The ROC of anti-BIRC5 autoantibody in ANHCC. (F) The bar chart of positive and negative rates in ANHCC and NC. The cut off value was 0.351 which is based on the 90% percentile of NC in Nanchang.

**Table 2 The diagnostic performance of the validation cohort in Zhengzhou and Nanchang.**

| | AUC (95% CI) | P | Se (%) (95% CI) | Sp (%) (95% CI) | PPV (%) | NPV (%) | Accuracy (%) |
|---|---|---|---|---|---|---|---|
| Zhengzhou | 0.733 [0.642–0.082] | <0.0001 | 28.1 [18.1–40.8] | 93.0 [83.3–97.2] | 80.0 | 56.4 | 60.5 |
| Nanchang | 0.745 [0.614–0.876] | 0.0017 | 39.3 [23.6–57.6] | 89.3 [72.8–96.3] | 78.6 | 59.5 | 64.3 |

**Note:**
Cut off: 0.351 was determined by the 90% percentile of Nanchang NC. Se, sensitivity; Sp, specificity; PPV, positive predictive value; NPV, negative predictive value.

depicting the predictive potential of genes at different time intervals (1, 3, and 5 years). Importantly, all these curves exhibit promising predictive capabilities, with the highest AUC value observed at 1 year (0.719; 95% CI [0.657–0.777]).

## Correlation analysis between BIRC5 expression and immune cell infiltration in HCC

Immunoinfiltrating cells play a crucial role in the initiation and progression of tumors. The relationship between BIRC5 and infiltrating immune cells within the immune microenvironment of HCC was examined using an online database (http://timer.cistrome.org/). The Fig. 7 revealed that BIRC5 expression exhibited significant associations with macrophages, CD4+T cells, CD8+T cells, neutrophils, B cells

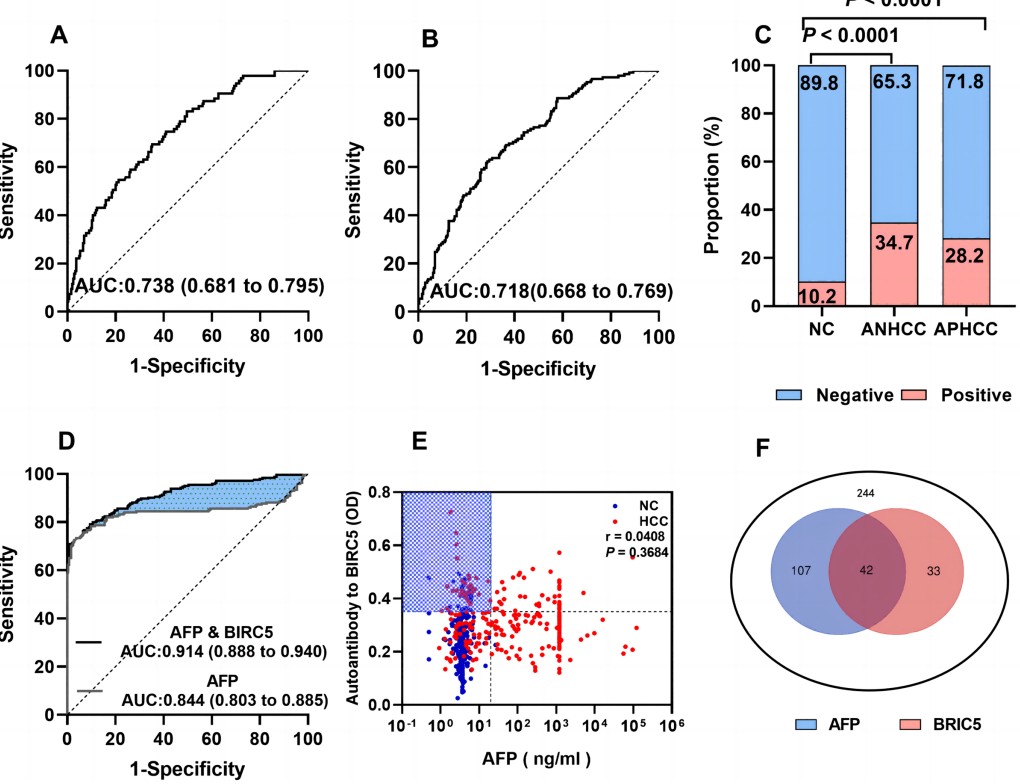

**Figure 4 Evaluation of the diagnostic performance of anti-BIRC5 autoantibodies.** (A and B), ROC curves of ANHCC and APHCC in the evaluation cohort. (C) Positive rates of anti-BIRC5 autoantibody in different groups while the cutoff value was 0.351 determined by the 90% percentile of Nanchang NC. (D) ROCs of AFP and logistic combined AFP and anti-BIRC5 autoantibody in the evaluation cohort. (E) The relationship of AFP and anti-BIRC5 autoantibody. (F) The Venn diagram of AFP and anti-BIRC5 autoantibody for HCC diagnosis.

**Table 3 Performance of the diagnostic performance of anti-BIRC5 autoantibody and AFP in evaluation phase.**

| | AUC (95% CI) | P | Cut off | Se (%) (95% CI) | Sp (%)(95% CI) | PPV (%) | NPV (%) | Accuracy (%) |
|---|---|---|---|---|---|---|---|---|
| Anti-BIRC5 autoantibody | 0.726 [0.682–0.770] | <0.0001 | 0.351 | 30.7 [25.3–36.8] | 89.8 [85.3–93.0] | 75.0 | 56.4 | 60.3 |
| AFP | 0.844 [0.803–0.885] | <0.0001 | 20 | 61.1 [24.8–67.0] | 99.6 [97.7–100.0] | 99.3 | 71.9 | 80.3 |
| Anti-BIRC5 autoantibody & AFP | 0.914 [0.888–0.940] | <0.0001 | 0.5 | 77.5 [71.8–82.3] | 91.8 [87.7–94.6] | 90.4 | 80.3 | 84.6 |

**Note:**
Cut off: 0.351 was determined by the 90% percentile of Nanchang NCs, Se, sensitivity; Sp, specificity; PPV, positive predictive value; NPV, negative predictive value.

and myeloid dendritic cells. These findings suggest a positive association between the presence of infiltrating immune cells and the expression of BIRC5 in HCC ($P < 0.05$).

## Correlation analysis of BIRC5 expression in relation to DNA methylation and gene mutation

Generally speaking, there exists a negative correlation between DNA methylation levels and gene expression levels. Based on the DiseaseMeth database (http://bio-bigdata.hrbmu.edu.cn/diseasemeth/analyze.html), it has been observed that in HCC patients, the DNA

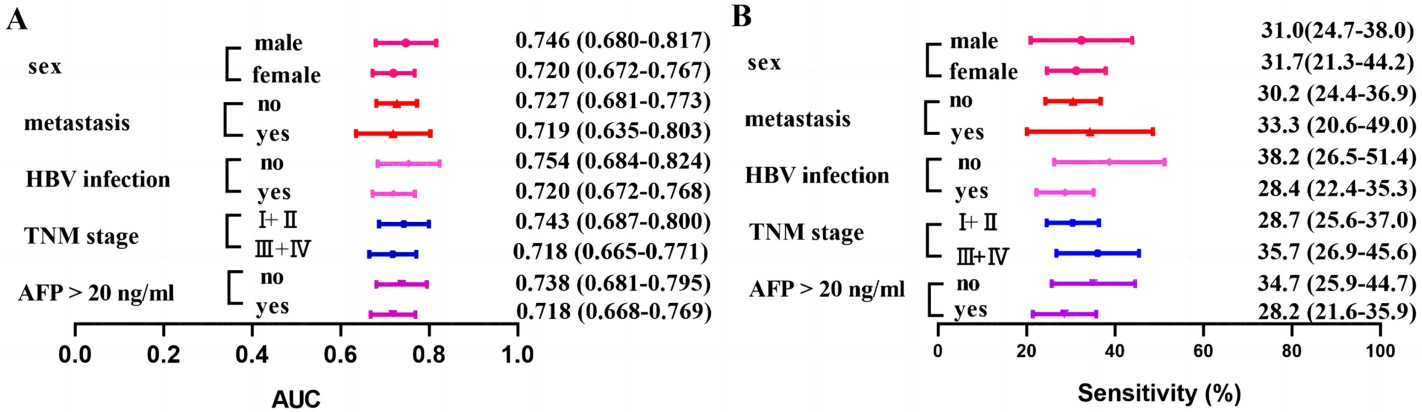

**Figure 5 Clinical AUC and sensitivity of anti-BIRC5 autoantibodies in different subgroups.** (A) The AUC of the anti-BIRC5 autoantibody in different subgroups. (B) The sensitivity of the anti-BIRC5 autoantibody in different subgroups.

**Table 4 Diagnostic performance of anti-BIRC5 autoantibody in different subgroups of evaluation phase.**

|  | AUC (95% CI) | P | Se (%) (95% CI) | Sp (%) (95% CI) | PPV (%) | NPV (%) | Accuracy (%) |
|---|---|---|---|---|---|---|---|
| AFP |  |  |  |  |  |  |  |
| APHCC (n = 149) | 0.718 [0.668–0.769] | <0.0001 | 28.2 [21.6–35.9] | 89.8 [85.3–93.0] | 62.7 | 67.2 | 66.4 |
| ANHCC (n = 95) | 0.738 [0.681–0.795] | <0.0001 | 34.7 [25.9–44.7] | 89.8 [85.3–93.0] | 56.9 | 78.0 | 74.3 |
| TNM stage, n (%) |  |  |  |  |  |  |  |
| I + II (n = 98) | 0.743 [0.687–0.80] | <0.0001 | 35.7 [26.9–45.6] | 89.8 [85.3–93.0] | 58.3 | 77.7 | 74.3 |
| III + IV (n = 92) | 0.718 [0.665–0.771] | <0.0001 | 28.7 [25.6–37.0] | 89.8 [85.3–93.0] | 59.7 | 70.4 | 68.6 |
| HBV infection, n (%) |  |  |  |  |  |  |  |
| Yes (n = 183) | 0.720 [0.672–0.768] | <0.0001 | 28.4 [22.4–35.3] | 89.8 [85.3– 93.0] | 67.5 | 62.6 | 63.5 |
| No (n = 55) | 0.754 [0.684–0.824] | <0.0001 | 38.2 [26.5–51.4] | 89.8 [85.3– 93.0] | 45.7 | 86.6 | 80.3 |
| Metastasis, n (%) |  |  |  |  |  |  |  |
| Yes (n = 39) | 0.719 [0.635–0.803] | <0.0001 | 33.3 [20.6– 49.0] | 89.8 [85.3– 93.0] | 34.2 | 89.4 | 82.0 |
| No (n = 205) | 0.727 [0.681–0.7730] | <0.0001 | 30.2 [24.4– 36.9] | 89.8 [85.3– 93.0] | 71.3 | 60.5 | 62.6 |

**Note:**
Cut off: 0.351 was determined by the 90% percentile of Nanchang NC. HCCs, hepatocellular carcinoma patients; NC, normal control.

methylation level is significantly reduced compared to that in normal samples (Fig. 8). The ICGC database (https://dcc.icgc.org/) was utilized to identify HCC tissues screened for mutations in the BIRC5 gene, as well as to determine the frequency of such mutations across five cohorts comprising 1,670 HCC tissues. As indicated by Table 5, among these donors, there were 41 cases affected by a total of 51 mutated genes. The frequency range for BIRC5 mutations varied from 0.25% to 5.43%, with an average mutation frequency of 2.62% across all five cohorts. These findings suggest a potential association between autoantibody emergence and mutations occurring within the BIRC5 DNA methylation and gene mutation.

## DISCUSSION

A comprehensive evaluation was carried out at multiple centers and stages to assess the significance of anti-BIRC5 autoantibody in the early detection of ANHCC. The results

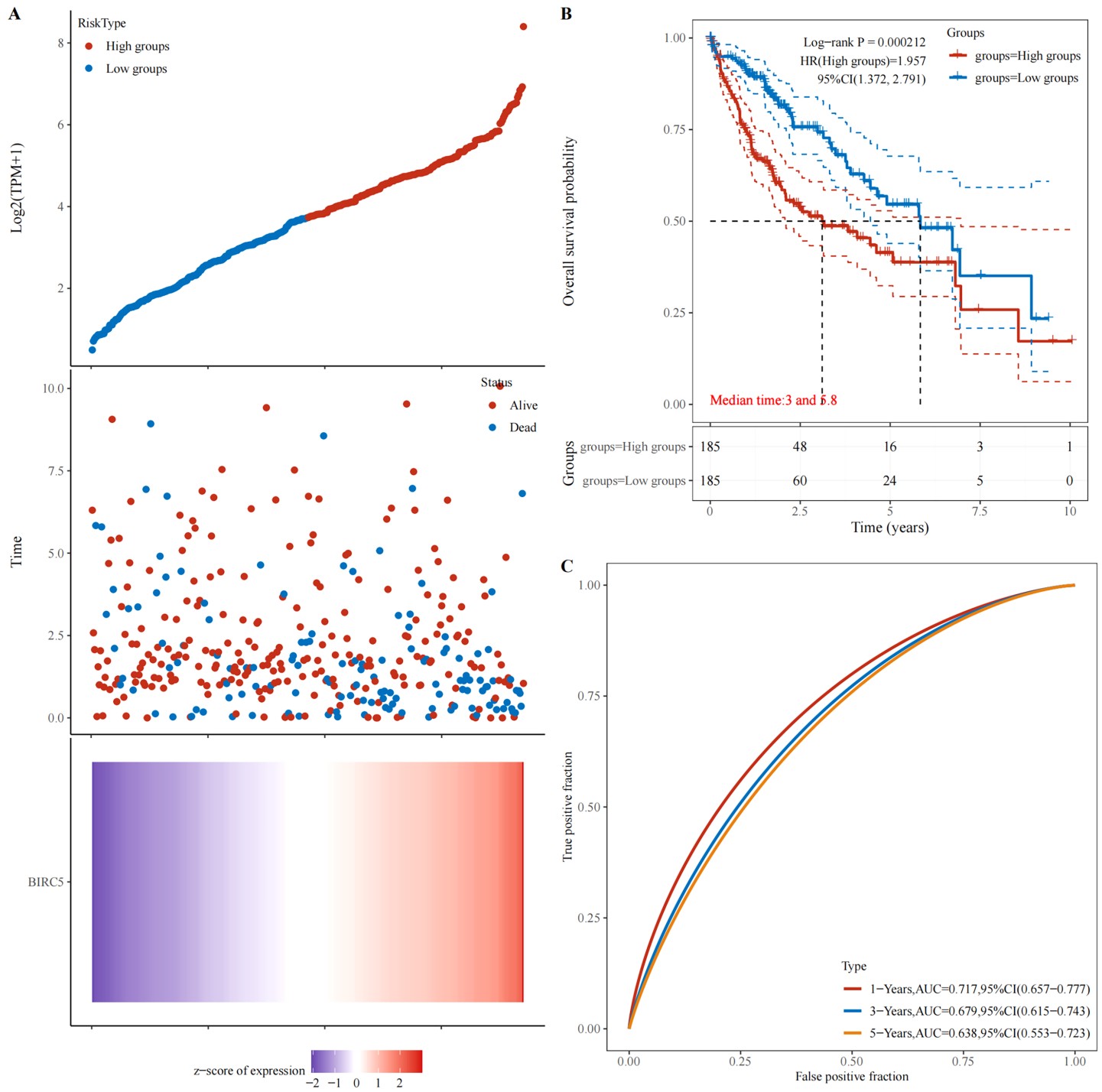

**Figure 6 Relationship between BIRC5 gene expression and prognosis.** (A) The relationship between gene expression and survival time and survival state in TCGA data. (B) K-M survival curve of BIRC5 genes in TCGA data. (C) ROC curve and AUC value of genes at different times.

revealed significantly elevated levels of anti-BIRC5 autoantibody in ANHCC patients compared to NCs. This discovery was further validated at two different centers before proceeding with an investigation involving a larger sample size. By combining anti-BIRC5

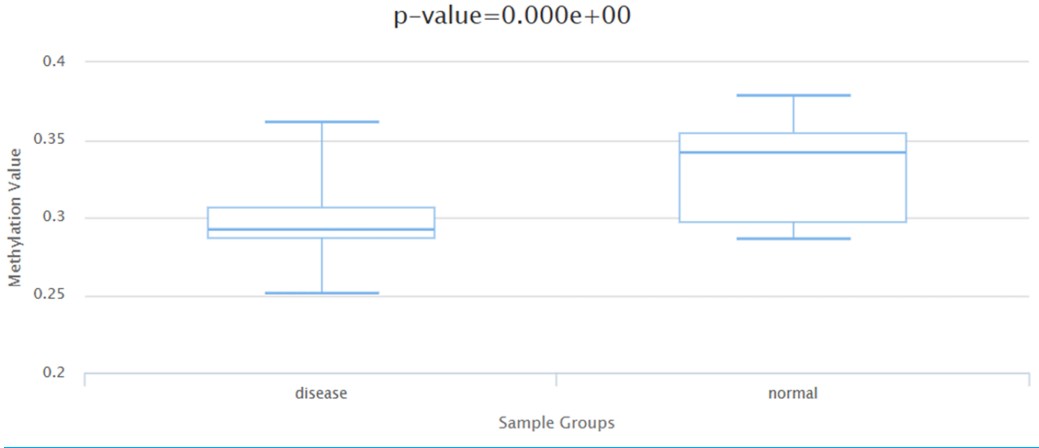

**Figure 7 The correlation between BIRC5 and immune cell infiltration in HCC.**

p−value=0.000e+00

**Figure 8 The DNA methylation level of BIRC5 in the DiseaseMeth database.**

**Table 5 BIRC5 gene mutations in the ICGC database.**

| Project | Site | Tumor type | Tumor subtype | Number of SSM-tested donors in the current project | Percentage of donors with the current gene | Mutations |
|---|---|---|---|---|---|---|
| Liver cancer—RIKEN, JP | Liver | Liver cancer | Hepatocellular carcinoma (Virus associated) | 258 | 0.054263566 | 14 |
| Liver cancer—FR | Liver | Liver cancer | Hepatocellular carcinoma (Secondary to alcohol and adiposity) | 252 | 0.023809524 | 12 |
| Liver hepatocellular carcinoma—TCGA, US | Liver | Liver cancer | Hepatocellular carcinoma | 364 | 0.008241758 | 3 |
| Liver cancer—NCC, JP | Liver | Liver cancer | Hepatocellular carcinoma (Virus associated) | 394 | 0.002538071 | 1 |
| Liver cancer—CN | Liver | Liver cancer | Hepatocellular carcinoma HBV-associated | 402 | 0.042288557 | 21 |

autoantibody with AFP, the resulting AUC value reached 0.914, indicating its potential as a biomarker for the timely diagnosis of HCC.

In line with our ongoing investigation, Meistere's study has also highlighted the potential of autoantibodies as indicators for tumors (*Meistere et al., 2017*). Furthermore, *Chen et al. (2010)*, *Yagihashi et al. (2005)*, and *Wu et al. (2014)* have independently identified anti-BIRC5 autoantibody as a promising marker for colorectal cancer, breast cancer, and oral cancer through single-stage validation. Although some studies suggesting the diagnostic significance of anti-BIRC5 autoantibody in HCC, their effectiveness in ANHCC remains unexplored (*Himoto et al., 2005*). In contrast to these prior investigations, our multi-center and multi-stage validation is the first to evaluate the diagnostic value of BIRC5 autoantibody specifically in ANHCC and its complementary role alongside AFP diagnosis for HCC.

During the process of tumor formation, alterations in both the quantity and quality of BIRC5 may initiate an immune response, resulting in the production of autoantibody in individuals with cancer (*Tyers & Mann, 2003*). Figure S1 illustrates the mRNA expression level of the BIRC5 gene obtained from the GEPIA database (http://gepia.cancer-pku.cn/). There is a notable distinction in BIRC5 mRNA expression between tumors and normal controls. A study reported that 70% (14 out of 20) hepatocellular carcinoma (HCC) tissues exhibited positive nuclear staining for BIRC5, while minimal detectable staining was observed in precancerous tissue using immunohistochemistry (*Ito et al., 2000*).

The effects of BIRC5 (Survivin) on the biological behavior of HCC are mainly reflected in the following aspects: (1) Inhibiting the apoptosis of HCC cells: BIRC5 protein binds to a variety of apoptotic proteases, inhibits the catalytic activity of caspase, and thus blocks the apoptosis process (*Shin et al., 2001*). (2) Promote the proliferation of HCC cells: Under the induction of bFGF, HCC cells express high levels of BIRC5 protein, which promotes the proliferation of cancer cells through activating the PI3K signaling pathway (*Sun et al., 2013*). (3) Induced tumor matrix angiogenesis: High-level BIRC5 expression in tumor tissues increases the level of β-catenin protein, enhances the transcriptional activity of

β-catenin/Tcf-Lef, and promotes the expression of VEGF, thus contributing to angiogenesis in tumor stroma (*Fernández et al., 2014*). (4) Molecular regulation of BIRC5 expression and function in HCC: It has been confirmed that the functional binding sites for STAT-3, KLF5, HIF-1α, Sp1, Rb, TCF4 and Egr1 are present in the BIRC5 promoter. Their bind mediates the activation of the BIRC5 promoter, thereby enhancing the expression level of BIRC5 (*Gritsko et al., 2006*; *Kim et al., 2003*; *Mityaev et al., 2010*; *Peng et al., 2006*). The mechanism of TAAbs production is complicated, and there are several theories: (1) defects in tolerance and inflammation; (2) altered protein structure; (3) overexpression of antigens; (4) cellular death mechanisms (*Zaenker, Gray & Ziman, 2016*).

On the whole, it can be inferred that elevated levels of anti-BIRC5 autoantibody found in HCC patients may have some plausible causes: First of all, overexpression of the BIRC5 gene leads to significant production of its protein which subsequently triggers corresponding anti-BIRC5 autoantibody. Secondly, Immune infiltration, gene methylation and gene mutation are all possible occurring in the BIRC5 gene locus may exert an influence on its expression, consequently impacting the production of anti-BIRC5 autoantibody. In addition, the amount of anti-BIRC5 autoantibody may also be affected by other regulatory mechanisms. This study shows that the expression of BIRC5 is positively associated with immune cell infiltration, indicating an "immune hot" tumor phenotype. Moreover, considering that the presence of autoantibody against BIRC5 likely reflects the immune response targeting BIRC5 antigen, it is suggested that BIRC5 autoantibody-positive cancers with a poor prognosis may exhibit a higher likelihood of favorable responses to immune checkpoint inhibitors.

The current study presents several notable strengths. Firstly, the diagnostic efficacy of anti-BIRC5 autoantibody has been validated across multiple centers research. Secondly, a substantial sample size was employed to assess the diagnostic potential in patients with ANHCC and APHCC, as well as its combination with AFP in detecting the whole HCC. Thirdly, the reason for the possible increase of autoantibody was speculated. Nevertheless, certain limitations exist. Firstly, this investigation did not encompass benign liver diseases and other tumor types, thus failing to further evaluate the specificity of anti-BIRC5 autoantibody for diagnosing HCC. Secondly, additional validation is required to ascertain the role of BIRC5 in initiating and advancing HCC. Next, we will continue to collect relevant samples for specific verification and explore the function of anti-BIRC5 autoantibody in the occurrence and development of HCC.

## CONCLUSIONS

Anti-BIRC5 autoantibody can be employed not only as a candidate biomarker for ANHCC, but also as a supplement to AFP in the diagnosis of whole HCC.

## ACKNOWLEDGEMENTS

The authors thank Professor Jianying Zhang for his invaluable support and insightful comments for this study.

## Funding

This research was funded by the Key Project of Tackling Key Problems in Science and Technology of Henan Province under Grant (No. 222102310066), the Project of Basic Research Fund of Henan Institute of Medical and Pharmacological Sciences (No. 2023BP0204-3) and The Science and Technology Project in Henan Province of China (No. 222102310408). The funders had no role in study design, data collection and analysis, decision to publish, or preparation of the manuscript.

## Grant Disclosures

The following grant information was disclosed by the authors:
Key Project of Tackling Key Problems in Science and Technology of Henan Province: 222102310066.
Project of Basic Research Fund of Henan Institute of Medical and Pharmacological Sciences: 2023BP0204-3.
Science and Technology Project in Henan Province of China: 222102310408.

## Competing Interests

The authors declare that they have no competing interests.

## Author Contributions

- Qing Li conceived and designed the experiments, performed the experiments, analyzed the data, prepared figures and/or tables, authored or reviewed drafts of the article, and approved the final draft.
- Haiyan Liu analyzed the data, authored or reviewed drafts of the article, and approved the final draft.
- Han Wang performed the experiments, authored or reviewed drafts of the article, and approved the final draft.
- Wenzhuo Xiong performed the experiments, authored or reviewed drafts of the article, and approved the final draft.
- Liping Dai conceived and designed the experiments, prepared figures and/or tables, and approved the final draft.
- Xiuzhi Zhang conceived and designed the experiments, prepared figures and/or tables, and approved the final draft.
- Peng Wang conceived and designed the experiments, prepared figures and/or tables, and approved the final draft.
- Hua Ye conceived and designed the experiments, prepared figures and/or tables, and approved the final draft.
- Jianxiang Shi conceived and designed the experiments, prepared figures and/or tables, and approved the final draft.
- Zhihao Fang analyzed the data, authored or reviewed drafts of the article, and approved the final draft.

- Keyan Wang conceived and designed the experiments, performed the experiments, analyzed the data, prepared figures and/or tables, authored or reviewed drafts of the article, and approved the final draft.

## Human Ethics

The following information was supplied relating to ethical approvals (*i.e.*, approving body and any reference numbers):

Ethics Committee of Zhengzhou University (ZZURIB 2019-001).

## Data Availability

The raw measurements are available in the Supplemental File.

## Supplemental Information

Supplemental information for this article can be found online at http://dx.doi.org/10.7717/peerj.17494#supplemental-information.

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
