# Peer review of "Anti-BIRC5 autoantibody serves as a valuable biomarker for diagnosing AFP-negative hepatocellular carcinoma"

_PeerJ, doi:10.7717/peerj.17494_

## Round 0.1 · original submission · Minor Revisions

Dear authors,

We kindly request that you carefully review the comments provided by the reviewers. Their valuable suggestions offer insights to enhance your manuscript. Incorporate their suggestions and carefully address all comments in your manuscript; This impact is particularly meaningful in the context of hepatocellular carcinoma research.

Reviewer 1 ·

Basic reporting

The language is clear and unambiguous, providing a structured overview of the research, methods, and conclusions. Sufficient background is given to understand the context and significance of the research. The introduction explains the limitations of current biomarkers and the potential of TAAbs in early cancer detection. No obvious typos were noticed in the provided text.

Experimental design

The methods are clearly described and appear to be reproducible. The research question is well-defined, relevant, and meaningful. The study aims to assess the diagnostic capability of anti-BIRC5 autoantibodies in detecting AFP-negative hepatocellular carcinoma (ANHCC), which addresses a clear gap in the current diagnostic methodology.

Validity of the findings

The manuscript presents a well-conducted study with significant findings that contribute to the field of cancer biomarkers. With some additional studies and data sharing, the strength and impact of the research could be further enhanced. Below are some of the recommendations to improve the manuscipt.

1) To strengthen the argument for the specificity of anti-BIRC5 autoantibodies, include a comparison with benign liver diseases and other tumor types.
2) Include functional studies to elucidate the role of BIRC5 in the initiation and progression of HCC.
3) If possible, include a longitudinal follow-up of patients to assess the prognostic value of anti-BIRC5 autoantibodies.
4) Consider expanding the diversity of the cohort to include different ethnicities and backgrounds to ensure the findings are generalizable.
5) Provide more detailed mechanistic insights into how BIRC5 expression and the immune response contribute to the production of autoantibodies.
6) Encourage the authors to share their data in a public repository to allow for independent verification of the results and to foster further research.

Reviewer 2 ·

Basic reporting

Basic reporting
Clear, unambiguous, professional English language used throughout.
Yes, the manuscript was written thoroughly in a clear and professional English language.
Intro & background to show context.
Introduction and background of the manuscript was very much relevant to the title of the research topic.
Literature well referenced & relevant. Y
Yes the literature was well referred throughout the manuscript and it is very much relevant to the study.
Structure conforms to PeerJ standards, discipline norm, or improved for clarity.
Yes, the study very much within the Peer J standards.
Figures are relevant, high quality, well labelled & described. Raw data supplied (see PeerJ policy).
Yes, the figures are relevant, high quality, well labelled and raw data was provided as per the journal policy.

Experimental design

EXPERIMENTAL DESIGN
Original primary research within Scope of the journal.
Yes the study falls very much within the scope of the journal.
Research question well defined, relevant & meaningful.
Research question is very much well defined and meaningful.
It is stated how the research fills an identified knowledge gap.
Rigorous investigation performed to a high technical & ethical standard.
Yes, validation of the findings were done ethically and with high quality.
Methods described with sufficient detail & information to replicate.

Validity of the findings

Conclusions are well stated, linked to original research question & limited to supporting results.

Reviewer 3 ·

Basic reporting

no comment

Experimental design

no comment

Validity of the findings

1) There might be a mistake in Table 1, where "APHCC" in Cohort 2 could be a typo for "ANHCC.
2)In addition to AFP, the usefulness of another liver cancer marker known as des-gamma-carboxy pro-thrombin (DCP) is recognized. However, it is not discussed. What do the authors think about this?
3)The authors have noted that the expression of BRICS5 correlates with immune cell infiltration, suggesting an "immune hot" tumor phenotype. Furthermore, considering that the autoantibodies against the cancer antigen evaluated in this study likely reflect the extent of the immune response against BRICS5, it is suggested that BRICS5 self-antibody-positive cancers with poor prognosis may have a higher likelihood of responding effectively to immune checkpoint inhibitors. Please elaborate on this point in the discussion section.

---

## Round 0.2 · accepted · Accept

The authors have conducted excellent work on this manuscript, incorporating feedback from reviewers to enhance the clarity and depth of the content. They have successfully elucidated the role of Anti-BIRC5 autoantibody role in the hepatocellular carcinoma, providing a detailed and comprehensive explanation of this complex biological process. This manuscript now offers a clearer and more insightful understanding of the mechanisms at play, thanks to the thoughtful integration of the reviewers' suggestions.